# Liebenberg syndrome severity arises from variations in *Pitx1* locus topology and proportion of ectopically transcribing cells

Olimpia Bompadre [1,2], Raquel Rouco [1,2], Fabrice Darbellay [1,2], Antonella Rauseo[1,2], Fanny Guerard-Millet[3,4], Claudia Gentile[3,5,6], Marie Kmita [3,4,5] & Guillaume Andrey [1,2] ✉

Enhancer hijacking, a common cause of gene misregulation linked to disease, occurs when non-matching enhancers and promoters interact ectopically due to genetic alterations. While the concept of enhancer hijacking is well understood, the reasons behind the variation in phenotypic severity remain unexplored. In this work, we expand on the ectopic activation of the hindlimb-specific transcription factor *Pitx1* by one of its own enhancers, *Pen*, in forelimb tissues that causes the Liebenberg syndrome. Using a series of inversions and relocations we show that reduction in *Pitx1-Pen* relative genomic positioning leads to increased proportions of *Pitx1* forelimb-expressing cells and more severe phenotypical outcomes. We demonstrate in ectopically expressing cells that the *Pitx1* locus assumes an active topology and that its promoter generates consistent transcription levels across different alleles. Finally, we show that changes in 3D chromatin structure and enhancer-promoter contacts are not the result of *Pitx1* transcription.

The restriction of enhancer-promoter contacts is a fundamental feature of gene regulation. This was shown to be mediated by domains of preferential interactions called topologically-associating domains (TADs). Indeed, TADs foster high internal chromatin interactions while reducing interactions with external regions. Biophysically, TADs are believed to be formed by a process called loop extrusion where cohesin molecules extrude chromatin until reaching CTCF which induces a temporary stalling of the process[1,2]. Changes in CTCF binding therefore impact the 3D architecture of loci and enhancer-promoter contacts[3]. Moreover, tissue-specific chromatin interactions can actively control enhancer-promoter communications in a spatiotemporally-defined manner, enabling the activation of associated genes[4–7].

Alterations in this organized process can lead to the wrongful connection between non-matching enhancers and promoters, leading to gene de-repression and expression in ectopic tissues, in a process named "enhancer-hijacking". In particular, structural variants (SVs) that impact the topological organisation of loci have been shown to lead to congenital malformations in such a way[8–10]. Although the *patho*-mechanism of SV-induced enhancer-hijacking has been documented across numerous loci, these accounts often overlook the influence of variations in SV breakpoints on disease outcomes or severity[11]. Furthermore, the precise relationship between distinct SVs and subsequent changes in the 3D genome architecture, chromatin modifications, and ectopic gene transcription is yet to be fully elucidated.

This is what happens at the *Pitx1* locus, where different SVs underlying the Liebenberg syndrome, a congenital malformation associated to a partial arm-to-leg transformation, are associated with variable morphological changes[12–15]. During normal development, the

¹Department of Genetic Medicine and Development, Faculty of Medicine, University of Geneva, Geneva, Switzerland. ²Institute of Genetics and Genomics in Geneva (iGE3), University of Geneva, Geneva, Switzerland. ³Genetics and Development Research Unit, Institut de Recherches Cliniques de Montréal, Montréal, QC H2W 1R7, Canada. ⁴Department of Medicine, Université de Montréal, Montréal, QC H3T 1J4, Canada. ⁵Department of Medicine, Division of Experimental Medicine, McGill University, Montréal, QC H4A 3J1, Canada. ⁶Present address: Dana-Farber Cancer Institute and Harvard Medical School, 450 Brookline Avenue, Boston, MA 02215, USA. ✉e-mail: guillaume.andrey@unige.ch

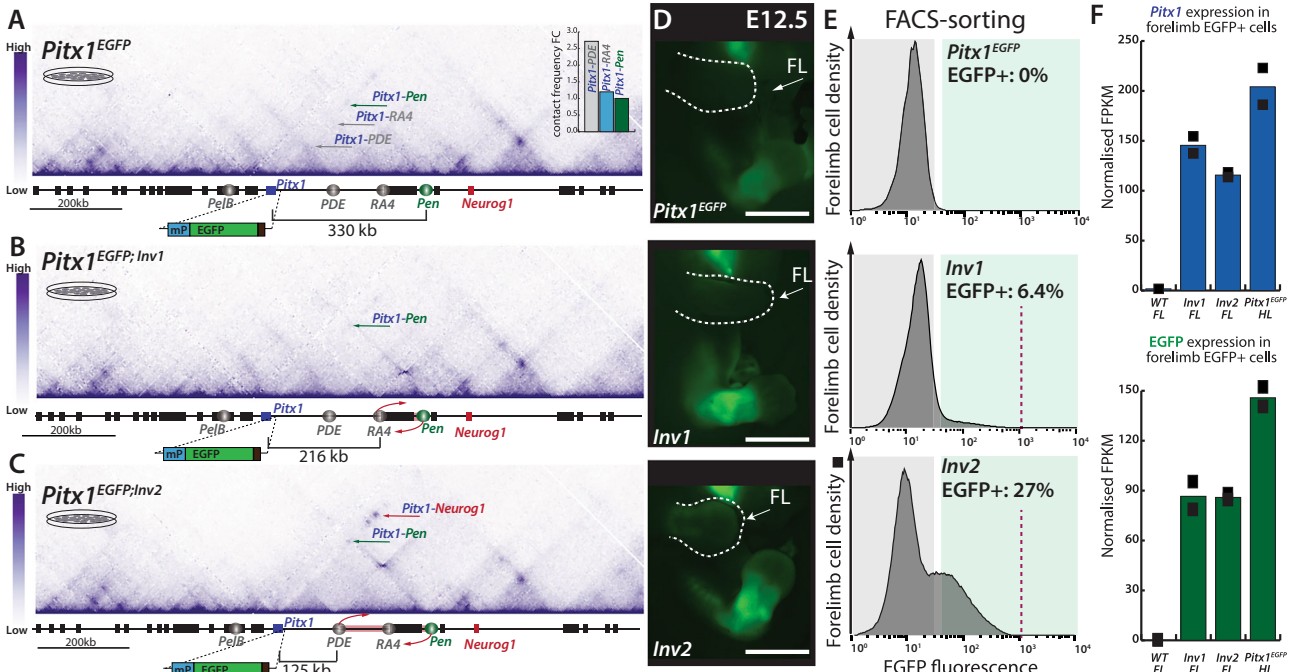

**Fig. 1 | Inversions at the *Pitx1* locus lead to increased mis-activation of the gene.** **A** C-HiC analysis of the *Pitx1* locus in *Pitx1^EGFP^* mESCs (n = 1). Upper right corner: quantification of interactions between *Pitx1* and *RA4/PDE/Pen*. **B** A 113 kb inversion *Pitx1^EGFP;Inv1^* (*Inv1*) that swaps the relative position of *Pen* and *RA4* shows a relative decrease in *Pitx1-Pen* interactions (n = 1). **C** A 204 kb inversion *Pitx1^EGFP;Inv2^* (*Inv2*) shows an overall increase of contacts between *Pitx1* and *Pen* (n = 1). **D** Fluorescence microscopy of *Pitx1^EGFP^*, *Inv1* and *Inv2* E12.5 embryos. Note the EGFP mis-expression in *Inv1* and *Inv2* developing forelimbs. Scale bars: 2 mm. Forelimbs (FL) are delineated with a dotted white line and indicated with a white arrow. In total, 43 *Pitx1^EGFP^* embryos, 19 *Inv1* and 9 *Inv2* E12.5 embryos have been analysed and showed reproducible patterns. **E** Histogram of EGFP signal and quantification of the proportion of mis-expressing cells. The grey and green areas show the delimitation of gating for EGFP- and EGFP+ cells, respectively, in the three alleles. The dotted red line in histograms indicates the upper limit of fluorescence. Each graph represents the catenation of 2-3 independent experiments. **F** Normalised FPKMs of EGFP and *Pitx1* in E12.5 wildtype bulk forelimbs, EGFP+ cells of *Inv1* and *Inv2* forelimbs and EGFP+ cells of *Pitx1^EGFP^* hindlimbs (Supplementary Data 2). Black squares indicate each biological replicate (n = 2). Note a plateau in *Pitx1* and EGFP expression in both inversions, which is significantly lower than in EGFP+ cells from *Pitx1^EGFP^* hindlimbs. Source data are provided as a Source Data file.

*Pitx1* gene is specifically expressed in developing hindlimb, and not in forelimbs, where it controls hindlimb outgrowth and differentiation into a leg[16–18]. So far, three limb enhancers have been identified at the locus: *PelB*, *RA4* and *Pen*[4,19]. Notably, another enhancer, *PDE*, has been described to contact the gene and as being strongly marked with H3K27ac in hindlimb, however, in reporter assays, the region only displays activity in the developing mandible[4,20]. Importantly, both *RA4* and *Pen* display a fore- and hindlimb activity when assayed in transgenic reporter approaches, and indeed, in the Liebenberg syndrome, the *Pitx1* gene gets *endo*-activated, i.e. ectopically activated by one of these two enhancers, *Pen*, in developing forelimbs[4]. This activation results from SVs that re-arrange the locus and generally bring *Pen*, normally located 400 kb away from *Pitx1*, in a closer genetic proximity to *Pitx1*. Patients with SVs that slightly reduce the *Pitx1-Pen* genetic distance show rather mild malformation features, yet, patients where *Pitx1-Pen* linear distance is strongly reduced display more severe ones (Supplementary Fig. 1, Supplementary Data 1)[12–15].

Here, we combine a previously developed *in-embryo* cell-tracing approach with engineered Liebenberg structural variants and *Pen* relocations to measure and isolate *Pitx1*-expressing cells in mouse forelimbs[21]. In this context, we explore how structural variants can cause different degrees of phenotypic manifestations by identifying their link to the proportion of ectopically expressing cells and transcriptional activities. Moreover, we investigate how de-repression or targeted activation of *Pitx1* can impact transcriptional activities and the locus topology.

## Results

### *Pitx1-Pen* relative genomic position affects the proportion of *Pitx1* ectopically expressing cells

To address how differential SVs breakpoints lead to gene mis-activation, we took advantage of the previously described *Pitx1^EGFP/EGFP^* (referred to as *Pitx1^EGFP^*) sensor allele that allows for the tracking and sorting of *Pitx1* active and inactive cells from developing tissues (Rouco et al., 2021). We re-engineered in the *Pitx1^EGFP^* background a previously published inversion leading to Liebenberg syndrome in mice: *Pitx1^EGFP;Inv1/EGFP;+^* (referred to as *Pitx1^EGFP;Inv1^*), as well a larger one *Pitx1^EGFP;Inv2/EGFP;+^* (referred to as *Pitx1^EGFP;Inv2^*) (Fig. 1A-C)[4]. These inversions place *Pen* at the positions of *RA4* and *PDE*, located 225 kb and 116 kb from *Pitx1*, respectively. At these locations, Capture-HiC (C-HiC) reveals that baseline interactions with *Pitx1* are stronger than *Pitx1-Pen* interactions in control *Pitx1^EGFP^* mouse embryonic stem cells (mESCs) (Fig. 1A). To measure how inversions perturb the locus poised 3D organisation, we performed C-HiC in *Pitx1^EGPP;Inv1^* and *Pitx1^EGFP;Inv2+/-^* mESCs. In *Pitx1^EGFP;Inv1^*, we observed a similar structure as in control mESCs (Fig. 1B). In contrast, in *Pitx1^EGFP;Inv2+/-^*, we observed several differences in the locus topology, with increased contact between *Pitx1*, *Pen*, and *Neurog1* (Fig. 1C).

We then derived *Pitx1^EGFP^*, *Pitx1^EGFP;Inv1^* and *Pitx1^EGFP;Inv2^* E12.5 embryos through tetraploid complementation and characterised forelimb EGFP fluorescence through microscopy and fluorescence activated cell sorting (FACS) (Fig. 1D, E; for gating strategy see Supplementary Fig. 2)[22]. We could measure in *Pitx1^EGFP;Inv1^* forelimbs 6.4% of EGFP+ expressing cells in contrast to 0% in *Pitx1^EGFP^* control (Fig. 1E).

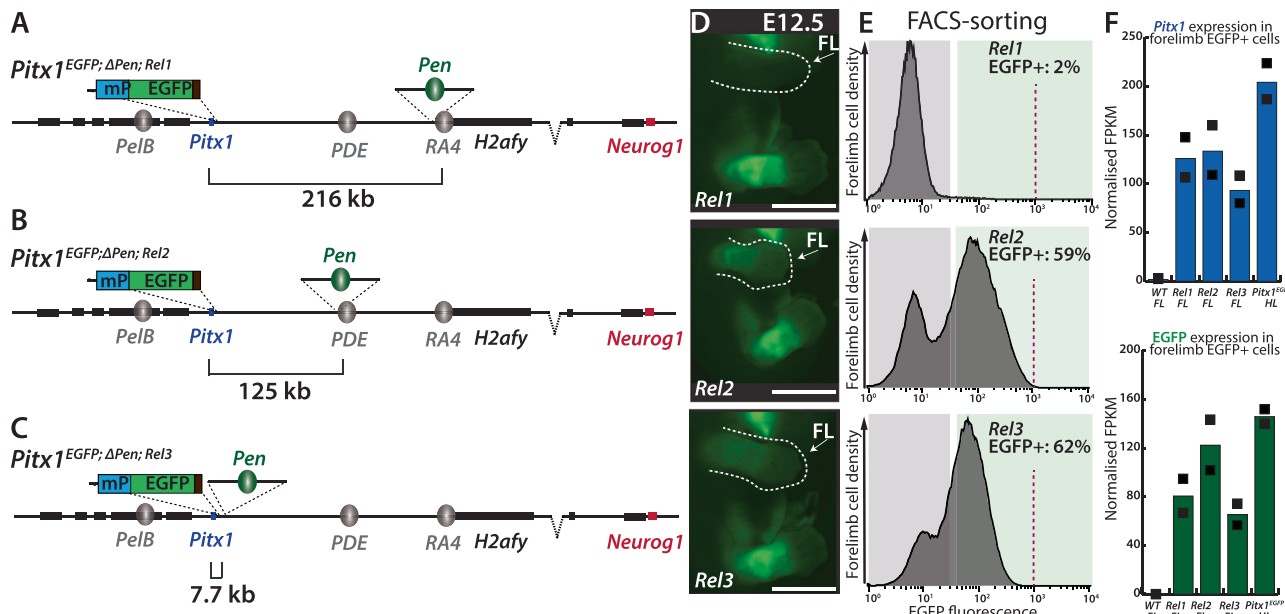

**Fig. 2 | Relocation of *Pen* through the locus ectopically activates *Pitx1*.**
**A** Illustration of *Pitx1^{GFP;ΔPen;Rel1}* (*Rel1*) where *Pen* is inserted, at *RA4*, 216 kb away to *Pitx1*. **B** Illustration of *Pitx1^{GFP;ΔPen;Rel2}* (*Rel2*) where *Pen* is inserted at *PDE*, 125 kb away from *Pitx1*. **C** Illustration of *Pitx1^{GFP;ΔPen;Rel3}* (*Rel3*) where *Pen* is inserted 7.7 kb away from *Pitx1*. **D** Fluorescence microscopy of *Rel1*, *Rel2* and *Rel3* E12.5 embryos. Note the mis-expression of EGFP and in developing forelimbs of E12.5 embryos. Fore-limbs (FL) are delineated with a dotted white line and indicated with a white arrow. Scale bars: 2 mm. In total, 17 *Rel1*, 5 *Rel2* and 8 *Rel3* E12.5 embryos have been analysed and showed reproducible patterns. **E** Histogram of EGFP signal and

quantification of the proportion of mis-expressing cells. The grey and green areas show the delimitation of gating for EGFP- and EGFP+ cells, respectively, in the three alleles. The dotted red line in histograms indicates the upper limit of fluorescence, each graph represents the catenation of 2-3 independent experiments.
**F** Normalised FPKMs of EGFP and *Pitx1* in E12.5 wildtype bulk forelimbs, EGFP+ cells of *Rel1*, *Rel2* and *Rel3* forelimbs and EGFP+ cells of *Pitx1^{EGFP}* hindlimbs (Supplementary Data 2). Black squares indicate each biological replicate (*n* = 2). Note the consistent *Pitx1* expression level between relocations, similar to inversion (See Fig. 1). Source data are provided as a Source Data file.

This number rose to 27% in *Pitx1^{EGFP;Inv2}* (Fig. 1E) suggesting that the variation in SV size can alter the proportion of cells ectopically expressing *Pitx1* in the forelimb.

Interestingly, we observed an upper limit of EGFP fluorescence in both inversions, suggesting that the abundance of EGFP in active cells was similar between alleles (Fig. 1E). To confirm that hypothesis, we measured transcription in EGFP+ cells of both *Pitx1^{EGFP;Inv1}* and *Pitx1^{EGFP;Inv2}* forelimbs using RNA-seq. We observed similar transcription levels of both *Pitx1* and *EGFP* across the two alleles (Fig. 1F, Supplementary Data 2). In fact, the ectopic transcriptional activity was only 1.5x lower than the one found in wildtype *Pitx1^{EGFP}* EGFP+ cells from hindlimbs (Fig. 1F). This minor difference might be the result of the heterozygous state of both inversions in the *Pitx1^{EGFP}* homozygous background, suggesting that the transcriptional activity per allele in forelimb EGFP+ cells is comparable to that in wildtype hindlimb EGFP+ cells. Yet, as the inverted intervals of both *Pitx1^{EGFP;Inv1}* and *Pitx1^{EGFP;Inv2}* contain CTCF sites (Supplementary Fig. 3A) and other *Pitx1* enhancers, the interpretation of the results can be confounding. Therefore, alternative approaches to solely measure the effect of the relocation of *Pen* were further developed.

**A series of *Pen* relocations induce varying proportions of *Pitx1*-expressing cells**
To rule out the positional effect induced by the inverted genomic interval, we devised a parallel approach where we solely re-mobilized the *Pen* enhancer itself in a *Pitx1^{EGFP/EGFP;ΔPen}* (referred to as *Pitx1^{EGFP;ΔPen}*) homozygous deleted background. Here, we inserted *Pen* at the same locations as in the inversions, at *RA4* (*Pitx1^{EGFP;ΔPen;Rel1/EGFP;ΔPen;+}*: *Pitx1^{EGFP;ΔPen;Rel1}*) and at *PDE* (*Pitx1^{EGFP;ΔPen;Rel2/EGFP;ΔPen;+}*: *Pitx1^{EGFP;ΔPen;Rel2}*) (Fig. 2A, B). Moreover, we also introduced *Pen* 7.7 kb upstream of the *Pitx1* promoter (*Pitx1^{EGFP;ΔPen;Rel3/EGFP;ΔPen;+}*: *Pitx1^{EGFP;ΔPen;Rel3}*), in a similar genetic distance (10.5 kb enhancer-

promoter distance) as the one found in the most severe case of Lie-benberg syndrome described (Fig. 2C, Supplementary Fig. 1)[15]. Of note, with each relocation reducing the genetic distance between *Pitx1* and *Pen*, there is also a consequent reduction in the number of CTCF binding sites separating these two elements (Supplementary Fig. 3B).

Similar to *Pitx1^{EGFP;Inv1}*, *Pitx1^{EGFP;ΔPen;Rel1}* E12.5 forelimbs showed 2% EGFP+ cells, suggesting that at this location the inversion and relocations bear a similarly mild transcriptional effect on *Pitx1* and the EGFP sensor (Fig. 2D, E). In contrast, in *Pitx1^{EGFP;ΔPen;Rel2}*, we measured 59% of EGFP+ forelimb cells and observed a clear bimodal EGFP signal distribution (Fig. 2D, E). This is twice the proportion observed when *Pen* was positioned at the same location in *Pitx1^{EGFP;Inv2}* forelimbs, where only 27% of cells were EGFP+ (Fig.1E). This difference suggests that the alterations in CTCF relative positioning and binding site directionality within the inverted interval might restrict the capacity of *Pen* to induce *Pitx1* in *Pitx1^{EGFP;Inv1}* forelimbs. Indeed, in contrast to *Pitx1^{EGFP;ΔPen;Rel2}*, the *Pitx1^{EGFP;Inv2}* allele causes the relocation and inversion of a *Pitx1*-con-vergent CTCF binding site at *PDE*, to the telomeric inversion break-point (Supplementary Fig. 3A-B). However, it cannot be excluded that the effect is also mediated by the interplay between *Pen* and *PDE*. Finally, in the E12.5 forelimb of the most proximal relocation, *Pitx1^{EGFP;ΔPen;Rel3}*, we observed a bimodal distribution of the EGFP signal and measured 62% of EGFP+ cells (Fig. 2D, E). Overall, the similar proportion of EGFP+ cells in *Pitx1^{EGFP;ΔPen;Rel2}* and *Pitx1^{EGFP;ΔPen;Rel3}*, shows that repositioning the enhancer either in the *PDE* region or a few kb upstream of the gene promoter induces a similar effect on *Pitx1* mis-activation (Fig. 2D, E, Supplementary Fig. 3B).

As inversions showed a similar transcription level between alleles in EGFP+ cells, we wanted to confirm this in the context of the relo-cations. We therefore performed RNA-seq in EGFP+ cells and found that *Pitx1* and EGFP expression is similar in all the active cells (Fig. 2F, Supplementary Data 2). Overall, this data shows that the ability of *Pen*

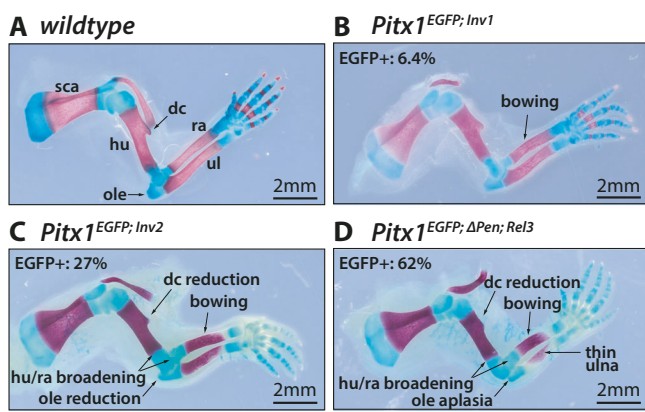

**Fig. 3 | Increasing proportions of *Pitx1* ectopically-expressing cells correlates with severity of skeletal defects. A** Alizarin red and alcian blue staining of wildtype E18.5 forelimbs. Black arrows pinpoint to sca: scapula, hu: humerus, dc: deltoid crest, ole: olecranon, ra: radius, ul: ulna. Alizarin red and alcian blue staining of mutants (**B**) *Pitx1EGFP;Inv1*, **C** *Pitx1EGFP;Inv2* and (**D**) *Pitx1ΔPen;Rel3* E18.5 forelimbs. Arrows point to dysplastic skeletal features: bowing of the radius and ulna, reduction of the deltoid crest, reduction of the olecranon, broadening of distal humerus and proximal radius and ulna, relative thinning of ulna. Number of independent limbs analysed and relative phenotypes are displayed in Supplementary Data 3.

to contact *Pitx1* defines the proportion of cells in which the gene will be ectopically activated, yet, it does not strongly affect *Pitx1* transcription level per allele.

## Increase in *Pitx1* ectopically expressing forelimb cells associate with worsened skeletal defects

As changes in *Pen* positioning lead to a different proportion of cells ectopically activating *Pitx1*, the phenotypic effect of these variations is unknown. To test whether an increase in affected cells is linked to a worsened phenotype, we analysed mutant skeletons of E18.5 embryos and scored forelimb malformations. We decided to compare wildtype to *Pitx1EGFP;Inv1*, *Pitx1EGFP;Inv2* and *Pitx1EGFP;ΔPen;Rel3* skeletons as these three precisely showed a progressive increase in EGFP+ cell proportions with 6.4%, 27%, and 62%, respectively. Weakly overexpressing forelimbs from *Pitx1EGFP;Inv1* resulted in a mild phenotype, specifically with a slight bowing of the radius and ulna (Fig. 3A, B, Supplementary Data 3). Notably, the same allele showed a stronger phenotype when bred to homozygosity and assayed in adult mice (Kragesteen, et al., 2018). *Pitx1EGFP;Inv2* forelimbs, where 27% of cells are EGFP+ at E12.5, showed more striking bowing of the radius and ulna (Fig. 3C, Supplementary Data 1, 3). Furthermore, we noted a significant reduction of the deltoid crest, a structure characteristic of the forelimb, accompanied by a mildly hypoplastic olecranon. Additionally, there was a noticeable broadening of the distal head of the humerus and the proximal head of the radius, a phenotype that aligns with previous descriptions in patients (Fig. 3C, Supplementary Data 1, 3). Finally, *Pitx1EGFP;ΔPen;Rel3* forelimbs, where 62% of cells are EGFP+ at E12.5, exhibited the most severe phenotype. This included the recurring bowing of the long zeugopodal bones, strong reduction of the deltoid crest, broadening of the distal humerus and proximal radius and notably, in all analysed *Pitx1EGFP;ΔPen;Rel3* skeletons, an aplastic or severely hypoplastic olecranon, a feature not observed in other alleles, but often in Liebenberg syndrome patients (Fig. 3D, Supplementary Data 1, 3). Finally, we observed a relative thinning of the ulna compared to its radius counterpart, in a similar way as the fibula is thinner than the tibia, underlining the arm-to-leg transformation. Most of the phenotypes scored across alleles were highly penetrant, with some variability in expressivity, even when only a limited number of cells were affected by *Pitx1* misexpression (Supplementary Data 3). Overall, our analysis shows that an increase of

*Pitx1* ectopically activating cells, and in fact a change in the balance between the proportion of active and inactive cells, has a positive correlation with the accumulation of defects in the developing forelimb skeleton.

## *Pitx1* forelimb *endo*-activation retains mesenchymal specificity

Liebenberg-associated SVs have been described to lead to arms assuming various skeletal and soft tissue features of legs[4,12,23]. To understand to what extent SV-induced *Pitx1* forelimb transcription resembles its normal hindlimb activity, we performed 10X single-cell RNA-seq (scRNA-seq) on stage-matched E12.5 *Pitx1Inv1/+* forelimbs and compared to wildtype fore- and hindlimbs. We selected this particular SV because it is a well-established model for Liebenberg syndrome and closely matches the rearrangement size of three SVs that have been independently identified in patients[4,12,14,21]. The first level of clustering revealed six main limb clusters: muscle, neuron, immune cells, epithelium, endothelium and mesenchyme (Supplementary Fig. 4A, Supplementary Data 4). We noticed that *Pitx1* expression was restricted to the mesenchyme in wildtype hindlimbs but also in *Pitx1Inv1/+* forelimb, although at lower expression levels (Supplementary Fig. 4A-B). We then subclustered the mesenchyme to obtain more definition to quantify *Pitx1* expression across sub-populations (Fig. 4A, Supplementary Data 4). Here, we identified nine mesenchymal populations comparable to the ones previously characterised in E12.5 limb mesenchyme[21]. Four clusters showed proximal identity: Proximal Proliferative Progenitors (**PPP**), Tendon Progenitors (**TP**), Irregular Connective Tissue (**ICT**) and Proximal Condensations (**PC**). An additional four clusters showed distal identity: Distal Proliferative Progenitors (**DPP**), Distal Progenitors (**DP**), Early Digit Condensations (**EDC**) and Late Digit Condensations (**LDC**). Finally, we identified a Mesopodium (**MS**) cell cluster, neither proximal nor distal. In *Pitx1Inv1/+* forelimbs, at least one *Pitx1* transcript could be detected in 34% of cells (compared to 2% in wildtype forelimbs), a higher percentage than the 6% of EGFP+ cells observed by FACS. This discrepancy can be attributed to the gating strategy, which excluded cells with weak GFP signals (See Figs. 1–2 and Supplementary Fig. 2), and the inherent sensitivity of both approaches. In wildtype hindlimbs and *Pitx1Inv1/+* forelimbs, we observed *Pitx1* expression in all mesenchymal clusters showing that the forelimb gain of expression occurred with a similar specificity than in hindlimbs. However, the variation in *Pitx1* expression between clusters was more pronounced in wildtype hindlimbs compared to *Pitx1Inv1/+* forelimbs (Fig. 4B, Supplementary Data 4). This observation indicates that the mesenchymal specificity of *Pitx1* expression is preserved in mutant forelimbs when compared to wildtype hindlimbs, albeit not to its full extent across mesenchymal subpopulations.

To assay whether these expression specificities are a general feature of *Pitx1 endo*-activation, we analysed the enrichment of marker genes in *Pitx1EGFP;Inv1*, *Pitx1EGFP;Inv2*, *Pitx1EGFP;ΔPen;Rel1*, *Pitx1EGFP;ΔPen;Rel2*, *Pitx1EGFP;ΔPen;Rel3* forelimbs and control *Pitx1EGFP* hindlimb EGFP+ cells compared to wildtype forelimbs. Generally, we observed homogenous marker gene enrichment among mutants, showing high similarity between EGFP+ cells (Supplementary Data 2). More specifically, all the EGFP+ populations showed a depletion of genes linked to non-mesenchymal cell identity (*Mrc1, Ttn, Krt14, Dlk2, Cldn5*) and an enrichment for mesenchymal markers (*Prrx1, Lhx9*) confirming that *Pitx1 endo*-activation specifically occurs in mesenchymal cell types (Fig. 4C). We also observed enrichment of proximal (*Shox2* and *Tbx15*), tendon (*Egr1*) and chondrogenic markers (*Sox9, Runx2*), corroborating the previous findings obtained from scRNA-seq. Furthermore, we also found that cells expressing *Pitx1* were enriched for cell division markers as *JunB* and *JunD* in line with the tissue outgrowth properties associated to *Pitx1*[21,24] (Fig. 4C). This shows the cell-specificity of *Pitx1 endo*-activation in forelimbs mirrors to a certain extent its physiological expression in wildtype hindlimbs.

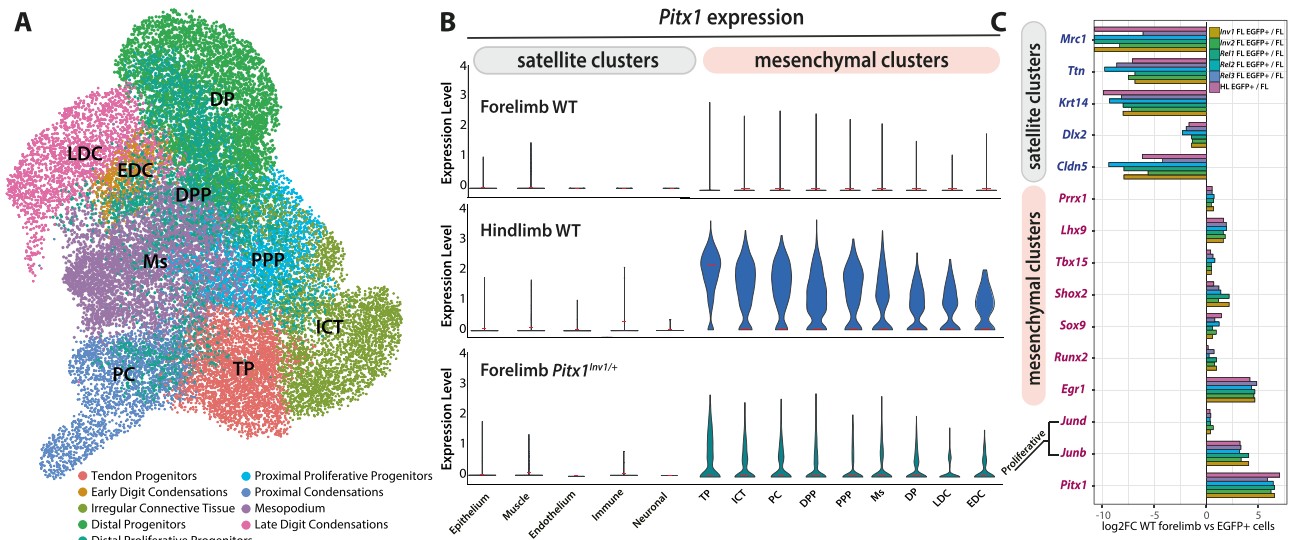

**Fig. 4 | *Pitx1* forelimbs ectopic expression is restricted to mesenchymal clusters. A** UMAP of mesenchymal cell clusters present in E12.5 wildtype fore- and hindlimbs and E12.5 *Pitx1^{Inv1/+}* forelimbs. **B** *Pitx1* expression by cell cluster in wild-type fore- and hindlimbs and *Pitx1^{Inv1/+}* forelimbs. Note the overall similarity of expression between wildtype hindlimbs and *Pitx1^{Inv1/+}* forelimbs. **C** Selected marker

genes enrichment across EGFP+ population of *Pitx1^{EGFP}* hindlimbs, as well as *Pitx1^{EGFP;Inv1}* (*Inv1*), *Pitx1^{EGFP;Inv2}* (*Inv2*), *Pitx1^{EGFP;ΔPen;Rel1}* (*Rel1*), *Pitx1^{EGFP;ΔPen;Rel2}* (*Rel2*), *Pitx1^{EGFP;ΔPen;Rel3}* (*Rel3*) forelimbs compared to wildtype bulk forelimbs. Source data are provided as a Source Data file.

Finally, to understand whether *Pitx1 endo*-activation can induce a wider hindlimb-like transcriptional program, we compared bulk *Pitx1^{EGFP;ΔPen;Rel2}* and *Pitx1^{EGFP;ΔPen;Rel3}* to wildtype forelimbs transcriptome. Here, we detected that the hindlimb-specific gene *Tbx4* was upregulated in mutant forelimbs, indicating that *Pitx1* expression could induce its transcription (Supplementary Data 5)[25]. We also noted an increase in cartilage and chondrogenesis related markers such as *Sox9*, *Foxc1* and *Gdf5* suggesting an increased chondrogenic program in mutant forelimbs (Supplementary Data 5)[18]. We then explored whether the *Pitx1^{Inv1/+}* forelimb scRNA-seq data showed a similar trend and found a comparable, though weaker, effect, with *Tbx4* and *Gdf5* significantly upregulated, while *Foxc1* and *Sox9* remained unchanged (Supplementary Fig. 4C). Altogether, these findings underline that *Pitx1 endo*-activation can establish, in the forelimb counterpart of hindlimb *Pitx1* expressing cell-types, features of hindlimb transcriptional programs.

## SV-induced *Pitx1 endo*-activation is linked to discrete topological changes

Hindlimb cells transcriptionally active for *Pitx1* adopt a fundamentally different 3D locus topology than their inactive counterparts[21]. Consequently, it is plausible that SVs-induced *Pitx1 endo*-activation leads to topological changes in transcriptionally active cells. To test this hypothesis, we initially performed C-HiC on *Pitx1^{EGFP;Inv1}*, comparing EGFP+ and EGFP- forelimb cells. We found that *Pitx1* contacts *Pen* as well as *PelB*, *PDE*, and *RA4* more frequently in EGFP+ cells than in EGFP- cells (Fig. 5A). Conversely, in EGFP- cells, the repressive contact between *Pitx1* and *Neurog1* was more prevalent than in EGFP+ cells (Fig. 5A). These differences are strikingly similar to those observed between hindlimb EGFP+ and EGFP- cells (Rouco et al., 2021), indicating that this inversion facilitates the formation of an active topology specifically in transcriptionally active cells.

We next investigated whether different active-inactive topologies would also be present in the other alleles described or if this was a specific feature of *Pitx1^{EGFP;Inv1}* forelimbs. Thus, we generated C-HiC maps of EGFP+ and EGFP- cells obtained from *Pitx1^{EGFP;Inv2}*, *Pitx1^{EGFP;ΔPen;Rel2}* and *Pitx1^{EGFP;ΔPen;Rel3}* forelimbs. In *Pitx1^{EGFP;Inv2}*, despite a higher proportion of *Pitx1*-expressing cells (See Fig. 1D-E), we observed fewer changes in interaction between EGFP+ and EGFP- cells. Here,

only the interaction between *Pitx1* and *Pen* was strongly increased in EGFP+ cells and, to a lesser extent, that between *Pitx1* and *PDE* (Fig. 5B). Similarly, in *Pitx1^{EGFP;ΔPen;Rel2}*, EGFP+ cells showed a clear gain of contacts between *Pitx1* and *PDE*, where the *Pen* enhancer is relocated, but not with other regions (Fig. 5C). These results consistently highlight strengthened *Pitx1-Pen* contact in transcriptionally active cells, suggesting that increased physical proximity is essential for transcription. Lastly, *Pitx1^{EGFP;ΔPen;Rel3}* EGFP+ and EGFP- forelimb cells exhibited limited topological changes (Fig. 5D). Here, due to the short 7.7 kb interval between *Pitx1* and *Pen*, the contact frequency between the two elements was very high in both active and inactive cells. Yet, we noted a relatively stronger contacts in EGFP- cells, a phenomenon already observed for active short-range regulatory contact (Fig. 5D)[26]. In conclusion, across the different gain-of-function alleles, we observe that fewer locus-wide topological changes associate with *Pitx1* transcription, i.e. in GFP+ vs GFP- cells, when *Pen* is brought closer to *Pitx1*.

We further explored whether changes in chromatin topology are associated with changes in *cis*-regulatory element activities by performing H3K27ac Chromatin Immunoprecipitation (ChIP-seq) in *Pitx1^{EGFP;Inv1}*, *Pitx1^{EGFP;ΔPen;Rel2}* and *Pitx1^{EGFP;ΔPen;Rel3}* EGFP+ cells, as well as in *Pitx1^{EGFP;Inv1}* EGFP- cells as a control (Supplementary Fig. 5). As expected in *Pitx1^{EGFP;Inv1}* EGFP- cells, we did not observe H3K27ac at *Pitx1* promoter, nor at several enhancers aside of *Pen*. In contrast, we observed a strong enrichment of H3K27ac at the *Pitx1* promoter in EGFP+ cells across alleles. In both *Pitx1^{EGFP;Inv1}* and *Pitx1^{EGFP;ΔPen;Rel2}* EGFP+ cells, there was an increase in H3K27ac coverage at *PDE*, a region interacting with *Pitx1* in both alleles. In *Pitx1^{EGFP;ΔPen;Rel3}* EGFP+ cells, only the region adjacent to the *Pen* relocation showed a clear acetylation signal (Supplementary Fig. 5). Finally, we also noted that in the two relocation alleles, the loss of *Pen* at its endogenous genomic location resulted in decreased H3K27ac spreading around it, while an increase around the *Pen*-relocated region was observed, showcasing the spreading potential of the histone mark. In summary, the increased chromatin contacts observed in C-HiC data involved regions marked by H3K27ac and therefore imply a type of homotypic interaction.

## Targeted activation of *Pitx1* does not induce topological change
In the context of SV-induced *Pitx1 endo*-activation, the relocation of the *Pen* enhancer associates with changes in transcriptional activity

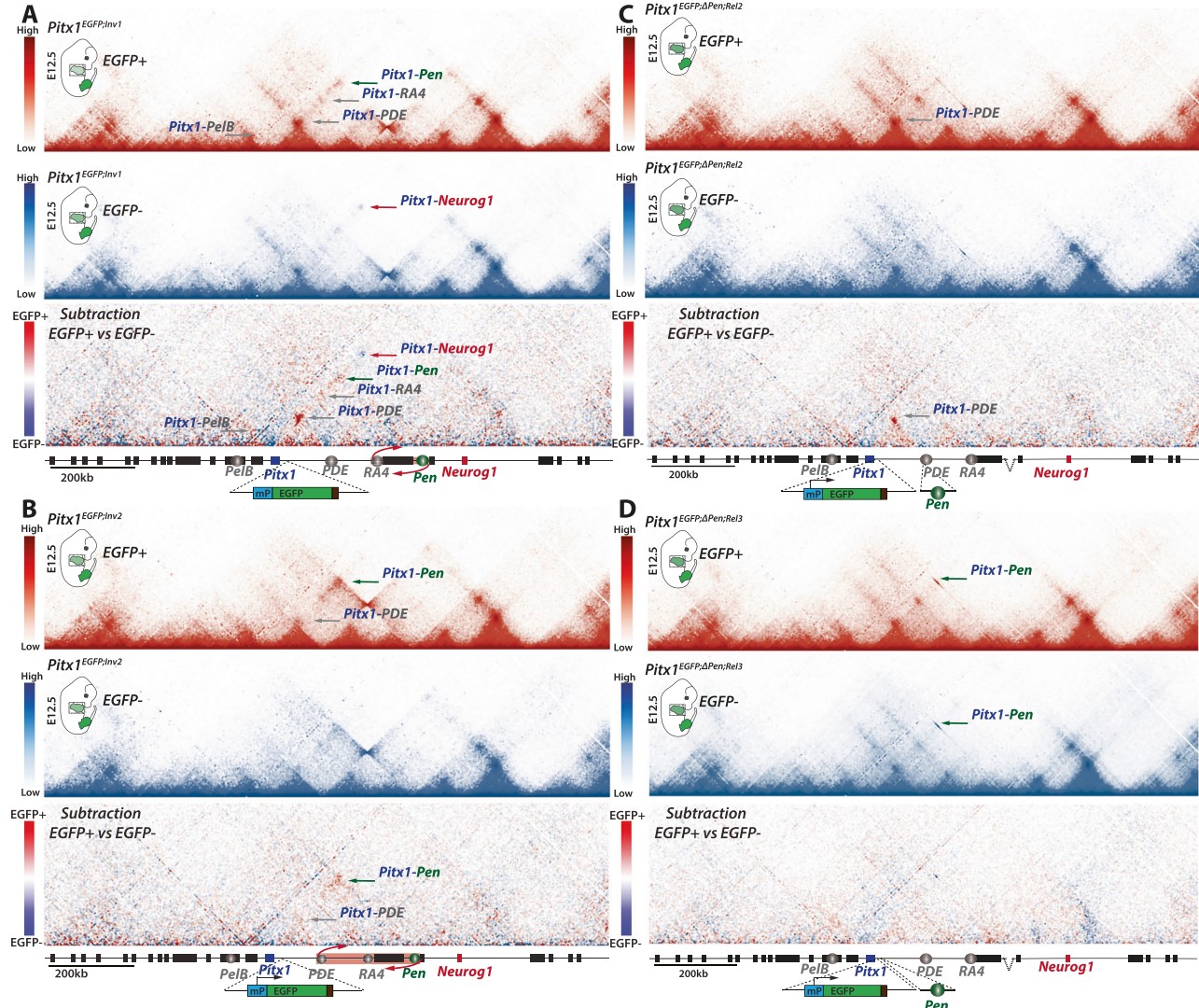

**Fig. 5 | Topological changes at the locus diminish as *Pitx1-Pen* contact probabilities increase.** C-HiC of the *Pitx1* locus in EGFP+ (red maps) and EGFP- (blue maps) cells from (**A**) *Pitx1^{EGFP;Inv1}* forelimbs, **B** *Pitx1^{EGFP;Inv2}* forelimbs, **C** *Pitx1^{EGFP;ΔPen;Rel2}* forelimbs and (**D**) *Pitx1^{EGFP;ΔPen;Rel3}* forelimbs. Darker red or blue bins indicate stronger interaction frequencies as shown on the scale bars. For each panel, the lowest map is a subtraction of the two above where preferential interactions in

EGFP+ cells are shown in red, while the ones in EGFP- cells are shown in blue. Contacts between *Pitx1* and *Pen* are shown with a green arrow, *Pitx1* contacts with *PelB*, *PDE* or *RA4* are shown with grey arrows, the *Pitx1-Neurog1* contact is shown with a red arrow. All subtraction scales were homogenized for comparison purposes. All experiments were performed once.

and genome topology (See Fig. 5 and Supplementary Fig. 5). Because both events occur in the same cells, it is unclear whether it is the transcription of the locus that induces the 3D topological changes or whether these occur independently. To assay whether ectopic activation of *Pitx1* is sufficient to induce changes in the locus topology, we first developed an in vivo dCas9-P300 activator targeted to the *Pitx1* promoter. To achieve specific expression of the activator in cell clusters permissive to *Pitx1* expression (See Fig. 4), we integrated the dCas9-P300 transgene preceded by a minimal promoter, as a sensor, upstream of the *Shox2* gene promoter to produce *Shox2^{dCas9P300/+}* mESCs (Fig. 6A). We selected *Shox2* because of its similar expression specificity with *Pitx1* in developing hindlimbs (correlation coefficient=0.577, *p*-value = 0.001, where the *p*-value is the probability for the correlation coefficient to be negative). To direct the dCas9-P300 activator to *Pitx1*, we integrated two sgRNAs that target the *Pitx1* transcriptional start site (TSS) at the *ColA1* locus to produce *Shox2^{dCas9P300/+};ColA1^{TSSsgR/+}* ESCs (Fig. 6A)[27].

We then derived E12.5 *Shox2^{dCas9P300/+};ColA1^{TSSsgR/+}* embryos using tetraploid aggregation[22]. Using RT-qPCR we could detect dCas9-P300 transcripts in forelimbs but not in the embryonic trunk, confirming the expression specificity of the sensor (Fig. 6B). Using RNA-seq, we measured *Pitx1* expression in *Shox2^{dCas9P300/+};ColA1^{+/+}* and *Shox2^{dCas9P300/+};ColA1^{TSSsgR/+}* forelimbs and could detect a 15-fold upregulation of the gene in the latter (Fig. 6C, Supplementary Data 6). As observed by whole mount in-situ hybridization (WISH), the expression pattern of *Pitx1* was localized to the proximal forelimb and reminiscent of *Shox2* expression in E12.5 forelimbs (Fig. 6D). Single-cell RNA-seq revealed that *Pitx1* was expressed in 9% of ^{dCas9P300/+};*ColA1^{TSSsgR/+}* forelimb mesenchyme compared to 2% of wild-type counterparts[21]. Moreover, we could generally observe that, *Pitx1* and *Shox2* expression domains colocalized in proximal clusters (*Pitx1-Shox2* correlation in the entire *Shox2^{dCas9P300/+};ColA1^{TSSsgR}* forelimb=0.441 *p*-value = 0.0005, where the *p*-value is the probability for the correlation coefficient to be negative Fig. 6E, Supplementary Fig. 6).

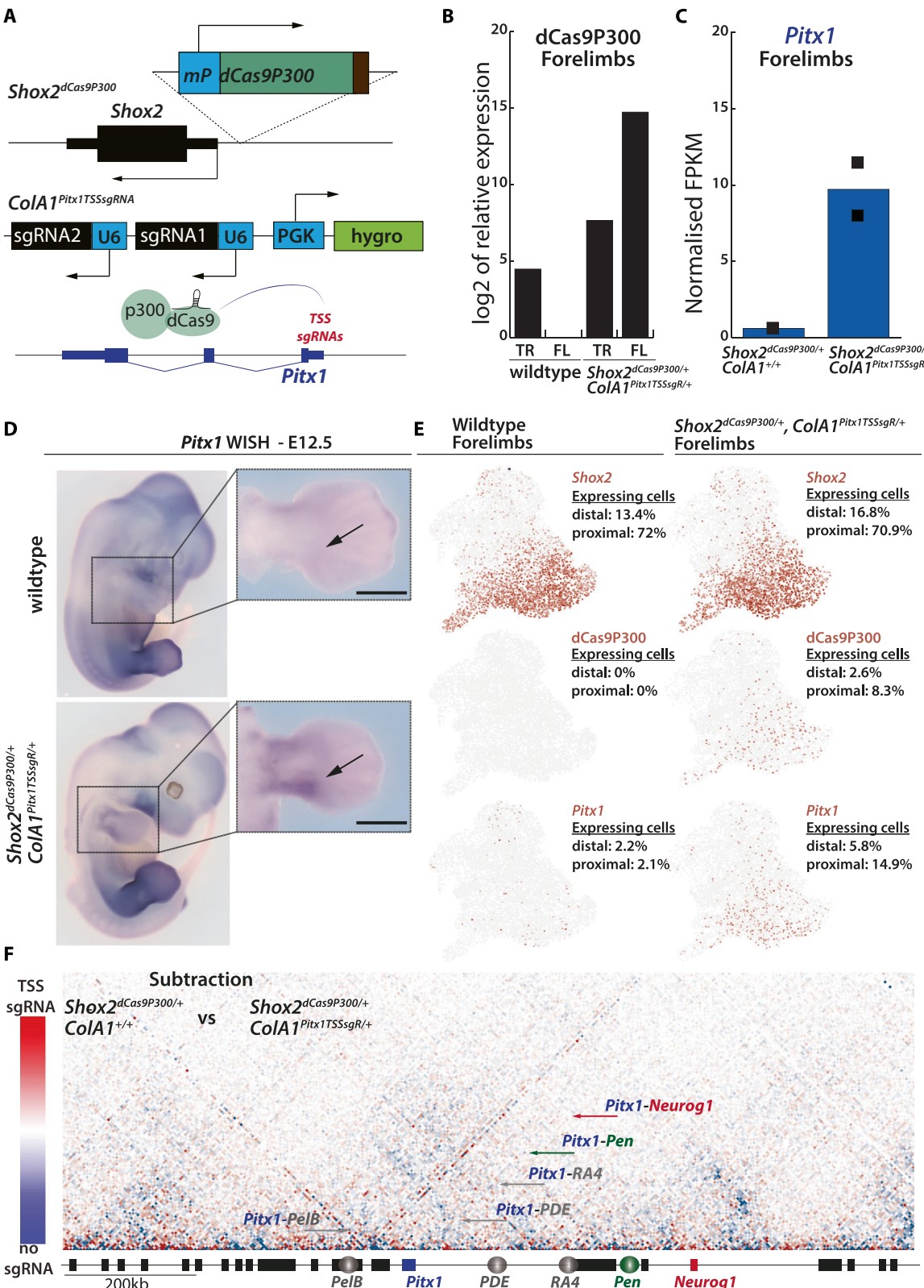

We next tested whether the gain of *Pitx1* transcription would elicit a change in 3D conformation of the locus. To enriched for *Pitx1* transcriptionally active cells, we micro-dissected E12.5 proximal forelimbs of *Shox2^dCas9P300/+;ColA1^TSSsgR/+* (14.9% of *Pitx1*-expressing cells) and *Shox2^dCas9P300/+;ColA1^+/+* (2.1% of *Pitx1*-expressing cells as defined in wildtype forelimb scRNA-seq) and performed C-HiC (Supplementary Fig. 7). By comparing *Shox2^dCas9P300/+;ColA1^TSSsgR/+* and *Shox2^dCas9P300/+; ColA1^+/+* C-HiC maps we did not observe change in locus interactions (Fig. 6F), suggesting that pronounced increases in topological contacts with *PelB*, *PDE*, *RA4*, and *Pen* are not driven by direct *Pitx1* activation. It is nevertheless possible that subtle changes may have been missed due to the small proportion of active cells in the sample studied.

**Fig. 6 | dCas9P300 induces *Pitx1* expression in forelimbs without topological changes. A** A dCas9-P300 cassette was inserted as a sensor upstream of the *Shox2* promoter, two sgRNAs to target dCas9 activity were integrated at the *ColA1* safe harbour locus through an FRT-mediated recombination. **B** RT-qPCR of dCas9P300 in wildtype and *Shox2^dCas9P300/+;ColA1^TSSsgR/+* E12.5 forelimbs (FL) and trunk (TR) tissues. The values represent a log2 fold change compared to wildtype forelimb that was set to 1. **C** Normalised *Pitx1* FPKMs in *Shox2^dCas9P300/+* and *Shox2^dCas9P300/+;ColA1^TSSsgR/+* (Supplementary Data 6). Black squares indicate each biological replicate (*n* = 2). Source data are provided as a Source Data file. **D** Whole Mount In-Situ Hybridization (WISH) of *Pitx1* in wildtype and *Shox2^dCas9P300/+;ColA1^TSSsgR/+* forelimbs. Note the proximal gain of *Pitx1* expression (black arrow; pattern observed in 4/4

embryos). Scale bars: 1 mm. **E** Individual UMAPS of scRNA-seq data from wildtype and *Shox2^dCas9P300/+;ColA1^TSSsgR/+* forelimbs showing the distribution of *Shox2*, *Pitx1* and dCas9P300 expressing cells as well as the respective percentage of expressing cells in proximal forelimb (proximal) and distal forelimb (distal). **F** Subtraction of *Shox2^dCas9P300/+;ColA1^TSSsgR/+* (TSS sgRNA) and *Shox2^dCas9P300/+;ColA1^+/+* (no sgRNA) E12.5 proximal forelimbs C-HiC maps. Contacts that are more frequent in TSS sgRNA are colored in red, and those more frequent in no sgRNA are colored in blue (See scale bar on the left). Contacts between *Pitx1* and *Pen* are shown with a green arrow, *Pitx1* contacts with *PelB*, *PDE* or *RA4* are shown with grey arrows, the *Pitx1*-*Neurog1* contact is shown with a red arrow. Note the absence of visible change. Corresponding individual C-HiC maps are shown in Supplementary Fig. 7 (*n* = 1).

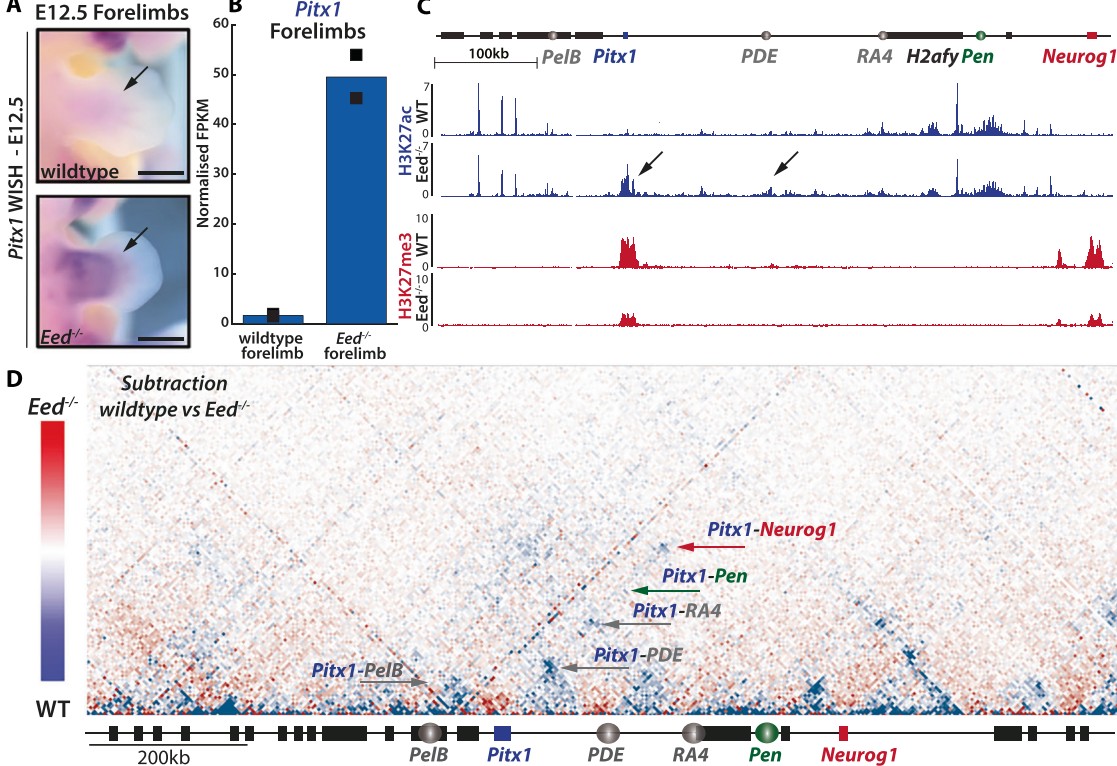

**Fig. 7 | PRC2 loss activates *Pitx1* transcription in forelimbs without increasing enhancer-promoter contacts. A** *Pitx1* Whole Mount In-Situ Hybridization (WISH) of E12.5 wildtype and *Prx1-Cre;Eed^flox/-* (*Eed^-/-*) forelimbs. Note the strong proximal gain of *Pitx1* expression (black arrow). Scale bars: 1 mm. **B** Normalised FPKMs of *Pitx1* in E12.5 wildtype and *Prx1-CRE;Eed^flox/-* (*Eed^-/-*) forelimbs. Black squares indicate each biological replicate (*n* = 2) (Supplementary Data 7). Source data are provided as a Source Data file. **C** ChIP-seq of H3K27ac (first two tracks) and H3K27me3 (two last tracks) show an accumulation of H3K27ac at the *Pitx1* locus (black arrows) in

proximal *Prx1-Cre;Eed^flox/-* (*Eed^-/-*) compared to wildtype (WT) forelimbs and an overall reduction of H3K27me3 signal. **D** Subtraction of wildtype and *Prx1-Cre;Eed^flox/-* (*Eed^-/-*) E12.5 proximal forelimbs C-HiC maps. Contacts more frequent in *Eed^-/-* are colored in red, and those more frequent in widtype are colored in blue (See scale bar on the left). Contacts between *Pitx1* and *Pen* are shown with a green arrow, *Pitx1* contacts with *PelB*, *PDE* or *RA4* are shown with grey arrows, the *Pitx1*-*Neurog1* contact is shown with a red arrow. Corresponding individual C-HiC maps are shown in Supplementary Fig. 8 (*n* = 1).

## Loss of PRC2 induces *Pitx1* forelimb transcription without increasing enhancer-promoter contacts

Because the targeted activation of *Pitx1* affected a limited proportion of forelimb cells, subtle topological changes could be missed. Therefore, we used a different approach to activate *Pitx1* transcription and asked whether removal of PRC2-mediated polycomb repression would be a more effective method. PRC2 is a multiprotein complex made of several subunits including the H3K27me3 reader EED which enables the spreading of the mark over chromatin domains[28]. Here we exploited a conditional *Eed* floxed allele combined to a full *Eed* knock out, and a limb-specific mesenchymal CRE driver (*Prx1-CRE;Eed^flox/-*) to assess the effect of its loss on both *Pitx1* transcription and locus structure[29–31].

Through WISH, we could observe a strong gain of *Pitx1* expression in proximal *Prx1-CRE;Eed^flox/-* E12.5 forelimbs (Fig. 7A). We then re-analysed RNAseq data and observed a 27-fold upregulation of *Pitx1* in

mutant forelimbs compared to wildtype littermates (Fig. 7B, Supplementary Data 7)[30]. As expected, in proximal E12.5 forelimb, a decrease in H3K27me3 could be detected throughout the locus (Fig. 7C)[32]. It is also interesting to note that despite the loss of H3K27me3 at *Neurog1*, the gene, unlike *Pitx1*, was not ectopically transcribed in forelimb cells, underlying cell-specificity as a requirement for mis-activation of genes (Fig. 7C). The decrease of H3K27me3 at *Pitx1* also coincided with the accumulation of the active H3K27ac mark at the gene promoter and *PDE* (Fig. 7C, black arrows)[30,32]. This shows that, at *Pitx1*, the removal of PRC2 repression results in the activation of the locus.

We then explored whether the loss of PRC2 repression results in a change of topological organisation of the locus (Supplementary Fig. 8). First, as expected from previous work[33], we observed a reduction of the *Pitx1-Neurog1* PRC2-associated contact in *Prx1-Cre;Eed^flox/-* proximal forelimbs compared to wildtype (Fig. 7D). However, similarly to the dCas9-P300 C-HiC data, we did not

observed a gain of interactions between *Pitx1* and its enhancers in *Prx1-Cre;Eed^flox/-* forelimbs (Fig. 7D). In fact, we observed a relative loss of the contacts with *PelB, PDE, RA4* and *Pen*, which suggests that PRC2 loss leads to a disorganisation of the locus topology. We concluded that loss of PRC2 leads to *Pitx1* activation independently from increased enhancer-promoter interactions.

## Discussion

In this work, we show that changes in the relative positioning between *Pitx1* and its *Pen* enhancer associate with a variable proportion of overexpressing cells in developing forelimbs. Within this active cell population, the levels of *Pitx1* expression do not increase with enhancer-promoter proximity but rather reach a conserved threshold of activation. This finding aligns with previous studies showing that enhancers primarily regulate transcription by increasing bursting frequency rather than bursting size[34,35]. Moreover, it suggests that once activation is achieved at the *Pitx1* locus it is done so at its full transcriptional potential where *Pitx1* promoter activity is saturated. However, the bimodal distribution of active and repressed cells contrasts with findings from other studies in mESCs, where enhancer repositioning led to a unimodal gene activation with varying transcriptional intensity, and that rarely left a proportion of cells in a repressed state[36,37]. This difference could stem from the heterogeneity of in vivo tissues or the specific nature of the *Pen/Pitx1* functional interactions.

Changes in *Pitx1-Pen* distance and its associated variation in the proportion of cells ectopically expressing *Pitx1*, but not in *Pitx1* transcription per allele, provide a mechanistic framework to account for the variation in Liebenberg syndrome severity among cases described so far. Here, we have shown that the more a SV reduces the *Pen-Pitx1* distance, and consequently the number of intermediate CTCF sites, the higher the proportion of forelimb *Pitx1* overexpressing cells will be and the more severe the skeletal defects. Similarly, patients with SVs inducing a short genomic distance and few intermediate CTCF binding between *Pitx1* and *Pen* displayed more severe malformations (Supplementary Fig. 1 and 9). In general, variability in rare disease severity has already been described in a few cases. For instance, several overlapping deletions at the *Epha4* locus, that induce rewiring of enhancers toward the *Pax3* gene, result in brachydactyly and variable hand defects[9]. Here, the proportion of cells affected by the *Pax3* overexpression in the distinct SVs could explain the variability in phenotypical outcome. In another reported case, different duplications at the *Ihh* locus, leading to variable increases of gene expression in developing limbs, were also shown to result in variable syndactyly phenotypes. Moreover, LacZ analysis of *Ihh* in the mutants indicated broadened expression domains of the gene, suggesting that increase in expression could be due to more cells ectopically activating *Ihh*[38]. Therefore, although our data provides a mechanism for variation in the Liebenberg syndrome, it could be applied to other syndromes linked to ectopic gene transcription.

In a previous study, we have shown that the homozygous loss of *Pen* did not result in a full *Pitx1* loss-of-function in hindlimbs, but in a 30% reduction of *Pitx1* transcription (Rouco et al., 2021). This hindlimb loss was mostly the result of a fraction of cells from all mesenchymal clusters, without further specificity, not displaying any *Pitx1* transcription. It was therefore hypothesised that *Pen* acts as a "support" enhancer enabling the robust *Pitx1* transcriptional initiation in the mesenchyme. In this perspective, other regions would act to provide more cell-type specificity, such as RA4 that was recently described as a chondrogenic enhancer[39]. This is similar to what happens during *endo*-activation, where *Pen* activates *Pitx1* in all forelimb mesenchymal clusters without further specificity. Together, *Pen*-dependent loss and gain of *Pitx1* expression pinpoint to the same role for *Pen*: to act as a *pan*-mesenchymal enhancer with the ability to trigger robust transcriptional onset at the *Pitx1* locus. As in hindlimbs other enhancers are required to further define *Pitx1* cell-type specific expression, it remains

to be shown whether other local enhancers, such as *RA4* which is also active in forelimbs, contribute to the final *Pitx1 endo*-activated expression in forelimbs.

By comparing the locus 3D topology in active and inactive cells, we observed that alleles driving *Pitx1* expression in a limited proportion of cells displayed the most extensive topological changes. Specifically, in the smallest inversion, *Pitx1^EGFP;Inv1*, multiple enhancer-promoter contact are observed in transcriptionally active cells involving interaction between *Pitx1* and *Pen* but also with *PelB, PDE* and *RA4*. This configuration is similar to the previously described stack configuration that occur, in fact, only in a fraction of *Pitx1*-expressing hindlimb cells[40]. In the other extreme, when the *Pen* enhancer was introduced directly upstream of *Pitx1*, in *Pitx1^EGFP;ΔPen;Rel3*, topologies were very similar between inactive and active cells. Together these data suggest that genetic configurations that reduce the searching space of the *Pitx1* promoter to find *Pen*, i.e. where the *Pen-Pitx1* contact is a very probable choice, are more likely to initiate transcription and therefore to result in *Pitx1* expression in a larger proportion of cells.

These changes in topology can be either the consequence of *Pitx1* transcriptional activation or an independent process required to activate and sustain *Pitx1* transcription. Here, our dataset can exclude the first possibility as the ectopic transcriptional activation of the *Pitx1* promoter via an exogenous dCas9-P300 activator or via the alteration of PRC2 activities did not result in a gain of enhancer-promoter contacts. This finding is consistent with similar results from targeted transcriptional activator approaches at other loci[26,41], reinforcing our results and raising further questions about how enhancer-promoter contacts are established in transcriptionally active cells. A recent study focusing on the *Pitx1* locus suggested that this process is controlled by changes in the loop extrusion process[40]. In fact, such changes might be directly influenced by the presence of RNA polymerase II at promoters and enhancers[42], a model that would account for the observed topological changes at *Pitx1*. However, this hypothesis seems unlikely, as neither ectopic transcription of *Pitx1* nor the endogenous activity of the *Pen* enhancer in forelimbs, both involving RNA polymerase II binding, is sufficient to drive these contacts. It is therefore plausible that other mechanisms related to *Pitx1* transcriptional activity influence loop extrusion. An alternative explanation could involve the formation of micro-compartments[43], independently of loop extrusion, facilitated by transcription factors. This model is supported by the observed loss of enhancer-promoter interactions in *Hoxc* genes or *Pitx1* knockout limbs[4]. In summary, our results demonstrate that transcription does not induce enhancer-promoter contacts at the *Pitx1* locus. Yet, such contacts, whether formed by loop extrusion or micro-compartments, could be essential to initially alter the state of the *Pitx1* promoter through its de-repression, and potentially to sustain its expression later on.

## Methods

### Animal procedures

Animal work performed in Geneva adheres to all relevant ethical regulations of the University of Geneva and follows procedures approved by the animal care and experimentation authorities of the Canton of Geneva, Switzerland (animal protocol numbers GE/89/19 and GE192A). Animal work performed at the Institut de Recherches Cliniques de Montréal (IRCM) was reviewed and approved by the IRCM animal care committee (protocols 2020-01 and 2021-04).

### Genetically engineered alleles

Engineered alleles using CRISPR/Cas9 technology were created in accordance with the methodology outlined in ref. 44. sgRNAs were designed using the Benchling software, selecting them based on predicted on-target and off-target scores. Detailed information on all sgRNAs and their corresponding genomic locations for CRISPR–Cas9 can be found in Supplementary Data 8. The sgRNAs were sub-cloned

into the pX459 plasmid from Addgene, with 8 μg of each vector utilized for the transfection of mESCs. Standard procedures for mESCs culture and genetic editing, were followed. The *Pitx1EGFP* mESCs clone used was previously described in ref. 20. Requests for transgenic G4 ESCs clones can be accommodated. C2 cells were engineered by frt-flippase system for recombination at the *ColA1* locus. C2 cells were transfected with 8 μg of recombination plasmid and 1 μg of FLP-recombinase plasmid. Transfection was done with Lipofectamine LTX-Plus (Thermo Fischer, 15338030) following the manufacture's transfection protocol. All primers used to characterise the alleles can be found in Supplementary Data 8.

## Culture of mESC cells and aggregation
G4 (129/sv x C57BL/6 F1 hybrid, male) and C2 (C57B/6 and 129svJae F1 hybrid, male) mouse ESCs, obtained from the Nagy laboratory (http://research.lunenfeld.ca/nagy/?page=mouse%20ES%20cells), were cultured on male and female CD1 feeders with complete growth medium supplemented with 1000 U/mL LIF (Murine Leukaemia Inhibitory factor ESGRO™(10^7 U/ml, Chemicon #ESG1107). Embryos were derived by tetraploid complementation from G4 and C2 ESCs. The mESCs were thawed and grown for two days. Donor tetraploid embryos were provided from in vitro fertilisation using the c57bl6J x B6D2F1 background. Aggregated embryos were transferred into CD1 foster females. Sample size calculation was not performed as there was no prior knowledge of percentages of GFP positive cells in forelimbs and numbers of embryos harvested per tetraploid aggregation. Therefore sample sizes were chosen relative to the numbers of GFP positive cells needed to perform experiments.

## Skeletal preparation
Skeletal preparation was performed as follows. E18.5 sacrificed foetuses were heat-shocked in H2O at 70 °C for 30" and skin and viscera were removed. This was followed by fixation in 100% EtOH at room temperature overnight and then in 100% acetone overnight at room temperature. Embryos were then stained in Alcian Blue (150 mg/l Alcian Blue 8GX Sigma-Aldrich) overnight at room temperature. Alcian Blue was removed and foetuses were washed with 100% EtOH and placed in Alzarin Red (50 mg/l Sigma Aldrich) in 0.2% KOH over two days. Finally, the remaining tissues were digested in 1% KOH with visual inspection and skeletons were stored in 0.2%KOH-30% glycerol for imaging and then long-term in 60% glycerol.

## Whole mount in situ hybridization
*Pitx1* WISH was performed on E12.5 embryos with a digoxigenin-labelled *Pitx1* antisense probe designed from a cloned antisense probe (PCR DIG Probe Synthesis Kit, Roche 11636090910). Experimental procedure followed the protocol outlined in ref. 4.

## Imaging
Embryos were imaged in PBS and skeletons in 0.2%KOH-30% glycerol on an Axio Zoom V16 (ZEISS) microscope. GFP laser exposure was set to 3000 ms.

## Preparation of single-cell limb suspension
E12.5 limb tissues were microdissected in cold PBS and pooled for processing. To maintain efficiency in downstream experiments, no more than 6 limbs were pooled together at a time. The tissues were dissolved in 400 μL Trypsin-EDTA and 40 μL 2.5% BSA (Sigma Aldrich, A7906-100G) over 12 min at 37 °C in a Thermomixer set at 1500 rpm, with a brief resuspension at the 6-min mark. Trypsin was quenched by adding 400 μL 2.5% BSA, and the homogenised tissue was passed through a 40μm cell strainer. An additional volume of 2.5% BSA was passed through to collect any remaining cells. The collected cells were then centrifuged 5' at 4 °C and 400 x g, followed by resuspension in 1%

BSA. If H3K27ac ChIP was planned as a downstream experiment, 5 mM NaButyrate (Sigma Aldrich, 303410) was added to the 1% BSA.

## Preparation for single-cell RNA-seq and library construction
Following the preparation of a single-cell limb suspension, cells were counted using an automated counter and resuspended to achieve a concentration of 1400 cells/μL. 50 μL of this suspension were provided to the iGE3 Genomic Platform for 10X Library Preparation. The platform performed library preparation for *Pitx1Inv1+/-* using the Chromium Single Cell 3′ GEM, Library & Gel Bead Kit v3.0 following the manufacturer's protocol. Libraries were pair-end sequenced on an Illumina HiSeq 4000 with approximately 8029 cells loaded on a Chromium Chip. For *Shox2dCas9P300/+;Cola1TSSsgR/+* library preparation was done using the Chromium Single Cell 3′ GEM, Library & Gel Bead Kit v3.1 following the manufacturer's protocol. Libraries were pair-end sequenced on an Illumina NovaSeq 6000 with approximately 10,141 cells loaded on a Chromium Chip.

## Cell sorting
Fluorescence-activated cell sorting (FACS) was employed to identify and sort distinct cell populations in this study, utilizing the Biorad S3 with GFP laser (excitation wavelength 488 nm). To eliminate debris from the analysis, FCC/FCS settings were established between 30/40 and 230/220. The viability stain Draq7 was employed to distinguish live cells, and standard protocols were applied to select for singlets. For each sample, a negative control tissue, the embryo's tails, was included to ensure the purity of the GFP- positive population. Moreover, the gating of GFP- positive populations was consistently applied across multiple experiments to ensure the selection of uniform populations and mitigate variability in GFP intensity over time. FlowJo™ Software was utilized for exporting the analysis in histogram format.

## Cell processing for ChIP-seq and C-HiC
After sorting, cells were suspended in 1% BSA and then centrifuged 5′ at 400 x g at 4 °C in a tabletop centrifuge. The supernatant was discarded, and cells were resuspended in 10% FCS/PBS before fixation at room temperature. For ChIP, 1% formaldehyde was used, and for C-HiC, 2% formaldehyde was applied, both for a duration of 10′ with rolling. Fixation was quenched by adding 1.45 M cold glycine, followed by centrifugation at 1000 x g, 8′, 4 °C. Cells were then resuspended in cold lysis buffer (10 mM Tris, pH 7.5, 10 mM NaCl, 5 mM MgCl2, 1 mM EGTA, Protease Inhibitor (Roche, 04693159001)). After 10′ of incubation on ice, fixed nuclei were isolated through a 3-min centrifugation at 1000 x g at 4 °C, followed by washing in cold 1 x PBS buffer (1000 x g, at 4 °C for 1 min). The PBS was removed, and nuclei were stored at -80 °C.

## Cell processing for RNA-seq and library preparation
For bulk limb analysis, two independent limbs were microdissected and snap-frozen at -80 °C for subsequent total RNA extraction using the RNEasy Mini Kit (QIAGEN, 74134) following protocol. RNA quantification was performed with Qubit 2.0 (LifeTechnologies) and the RNA Broad Range Assay (Q10210).

For GFP population studies, after sorting, at least two replicates of 2.5 ×10^5 cells were pelleted 5′ at 400 x g, 4 °C. After removal of 1% BSA, cells were snap-frozen at -80 °C for total RNA extraction. RNA extraction was carried out with the RNEasy Micro Kit (QIAGEN, 74004) following the manufacturer's instructions. Quantification was performed with Qubit and RNA High Sensitivity Assay (Q32852).

Library preparation and sequencing were conducted at the iGE3 Genomic Platform. RNA integrity was assessed with a Bioanalyzer (Agilent Technologies). The SmartSeq v4 kit (Clontech) was used for reverse transcription and cDNA amplification, following the manufacturer's instructions, with 5 ng RNA as input. Library preparation followed with a 200 pg cDNA input, using the Nextera XT kit (Illumina). Libraries were assessed by Tapestation and Bioanalyzer with a DNA

High Sensitivity Chip, 2 nM were pooled and sequenced on an Illumina NovaSeq 6000 sequencer using SBS TruSeq chemistry with an average of 35 million reads (single-end 50 bp) per library.

## RT-qPCR

RNA was extracted from forelimbs and trunks using the RNEasy Mini Kit (QIAGEN, 74134) and was converted into cDNA using the SuperScriptII RT kit (Invitrogen #18064-014), starting with 200 ng of RNA. The process utilized random hexamer primers (Thermo Scientific #S0142), DTT (Invitrogen Y00147), and RNaseOUT (Invitrogen 100000840). The amplification of cDNA was carried out with PowerUP™ SYBER Green Master Mix (Applied Biosystems A25742) and conducted on the QuantStudio 1 RT-PCR System from Applied BioSystems. The average CT values were standardized against the *Gapdh* housekeeping gene. The RT-qPCR primers for dCas9-P300 and *Gapdh* are detailed in Supplementary Data 8.

## Immunoprecipitation

Nuclei were sonicated to an average size of 200-500 bp fragments on a Bioruptor Pico Sonicator (Diagenode) for 8′ 30″ON/OFF cycles at 4 °C. Immunoprecipitation was performed with the α-H3K27Ac antibody (Diagenode C15410174) at a 1/500 dilution, 5 mM of Na-Bu was added to all buffers. Before sonication, magnetic beads were pre-cleared with 30 μL of Protein G beads (for H3K27ac – Invitrogen 10003D) and 0.25% BSA in PBS. After the addition of the antibody, the beads were left to rotate at 4 °C for at least 4 h. Unbound antibodies were removed, and following sonication, the chromatin was added to the beads and incubated rotating overnight at 4 °C. Unbound chromatin was then removed by seven washes in RIPA buffer and one in TE buffer. Chromatin was eluted and de-crosslinked overnight with the addition of 5 μL Proteinase K (10 mg/mL, Promega V3021). RNase A (4 μL, 10 mg/mL, ThermoFisher EN0531) treatment followed, and then phenol:chloroform:IAA extraction and precipitation. Chromatin was eluted in 50 μL $H_2O$.

## Library preparation and sequencing

Library preparation was performed by the iGE3 Genomic Platform. The Illumina ChIP TruSeq protocol was followed with a < 10 ng DNA input, and libraries were sequenced as 50 bp single-end reads with the Illumina NovaSeq 6000 sequencer. Libraries were validated on Tapestation and Qubit fluorimeter, pooled as 2 nM, and sequenced with TruSeq SBS chemistry.

## Capture-HiC and library preparation

C-HiC experiments were conducted as singlets using an average of $1 \times 10^6$ fixed nuclei for sorted cells and $3 \times 10^6$ mESC cells. The experiments adhered to the protocol outlined in Kragesteen et al., 2018, and Paliou et al., 2019. In this process, chromatin underwent digestion with the DpnII enzyme (1000U total; NEB, R0543M) at 37 °C overnight, supplemented with 20% SDS and 20% Triton X-100. Subsequent ligation was carried out with 100U of ligase in a 1.15% Ligation buffer (ThermoFisher, EL0012) at 16 °C for 4 h, followed by 30 min at room temperature. The decrosslinking step occurred overnight at 65 °C with the addition of 30 μL Proteinase K (10 mg/mL, Promega V3021). RNAse A treatment (30 μL, 10 mg/mL, ThermoFisher EN0531), 45′ at 37 °C, was followed by phenol:chloroform:IAA extraction and an overnight precipitation. After precipitation, the DNA pellet was reconstituted in 150 μL Tris pH7.5. Total DNA quantification was performed using the Qubit High Sensitivity DNA Assay (Q32851).

## Preparation of 3 C library and sequencing

Libraries were prepared by the iGE3 Genomic Platform. In brief, chromatin was sheared, and adapters were ligated following the manufacturer's protocol for Illumina sequencing (Agilent). Libraries underwent pre-amplification and hybridization on custom Sure Select beads spanning the chr13: 54,000,001–57,300,000 region, indexed for sequencing as 50 bp paired-end reads (Agilent). Once again, 2 nM of libraries were clustered for sequencing on an Illumina Novaseq 6000 with SBS TruSeq chemistry.

## Data analysis

**RNA-seq.**[45](Brawand et al. [45])RNA-seq reads were processed using CutAdapt v1.18 to trim low-quality bases and NextSeq sequencing adapters (-a CTGTCTCTTATACACATCTCCGAGCCCACGAGAC, quality cutoff -q30 and minimum length required -m15). Unstranded reads were mapped to the relevant GRCm39/mm39 custom genome filtered GTFs (see *Custom Genomes* section below) with using the STAR 2.7.2b mapper with settings allowing for accurate gene quantification (--outSAMstrandField intronMotif--sjdbOverhang '99' --sjdbGTFfile $gtfFile--quantMode GeneCounts--outFilterType BySJout--outFilterMultimapNmax 20 -- outFilterMismatchNmax 999--outFilterMismatchNoverReadLmax 0.04--alignIntronMin 20 -- alignIntronMax 1000000--alignMatesGapMax 1000000--alignSJoverhangMin 8 -- alignSJDBoverhangMin 1). Output BigWig files were displayed on the UCSC genome browser. Counts were compiled from STAR counts using R 3.6.2, and FPKM were computed through Cufflinks 2.2.1 using the filtered GTFs created for this study (--max-bundle-length 10000000 -- max-bundle-frags 100000000 -- multi-read-correct--library-type "fr-firststrand" --no-effective-length-correction -M MTmouse.gtf). The code is available on https://github.com/bompadreolimpia/Bompadre_etal_2024.

Count normalisation and differential expression analysis was done following published pipelines (https://github.com/lldelisle/rnaseq_rscripts). Briefly, normalized FPKM values were calculated by first determining coefficients extrapolated from a set of 1,000 housekeeping genes known for their stable expression as defined from the comparison of a series of RNA-seq (Brawand et al. [45]). The coefficients obtained were then applied to adjust the respective FPKM values. Differential expression analysis utilized the DEseq2 R package (version 1.38.3), with the Wald test for comparisons across samples and multiple test correction using the FDR/Benjamini-Hochberg test. Each analysis included two biological replicates per condition. Fold-enrichment of *Pitx1* and was calculated using DEseq2's normalization by size factor. *Custom Genomes.* For RNA-seq analysis, custom mm39 genomes were generated using STAR 2.7.2b, incorporating an additional chromosome to accommodate the custom sequences of EGFP and SV40 polyA tails. The gft file was modified to specify these sequences as coding genes and exons. Cell Ranger 6.1.2 was utilized for single-cell RNA-seq analysis, creating a custom mm39_dCas9P300 genome by adding an extra dCas9P300-containing chromosome and customizing the reference gtf file. The code is available on https://github.com/bompadreolimpia/Bompadre_etal_2024.

**ChIP-seq.** Reads from ChIP-seq data generated for this study were pre-processed withCutAdapt v1.18 to trim low quality bases and TruSeq adapters (-a GATCGGAAGAGCACACGTCTGAACTCCAGTCAC, -q30 and -m15). Reads were then mapped to the reference GRCm39/mm39 genome using Bowtie2 2.3.5.1 with default settings. Only reads with mapping quality score (MAPQ) of 30 or above were retained by filtering with SAMtools v1.10. For coverage and peak analysis reads were extended by 200 bp and processed with MACS2 v2.2.7.1 (--broad --nolambda --broad-cutoff 0.05 --nomodel --gsize mm --extsize 200 -B 2). Coverage normalization was performed by MACS2, normalized by the number of million tags used by MACS2. BedGraphToBigWig v4 was used to convert files into BigWig format for visualization in the UCSC browser. Datasets of CTCF ChIP-seq of mouse embryonic E11.5 forelimb (Andrey et al., 2017) and human fetal limbs day 58/59 (ENCODE, annotation file set ENCSR191WSJ) were visualized in the UCSC browser. The code is available on https://github.com/bompadreolimpia/Bompadre_etal_2024.

**Capture-HiC**. Capture-HiC data analysis was done as follows. Paired-end reads were mapped against the reference NCBI37/mm9 genome using Bowtie2 v2.3.4.2 for mapping of short reads. Filtering, de-duplication, and processing of valid and unique di-tag pairs were performed with HiCUP v0.6.1 with default parameters for configuration file, but adding Nofill: 1 as an additional parameter[46]. Valid and unique read pairs were then processed with Juicer Tools v1.9.9 to produce binned contact maps with MAPQ ≥ 30 and maps were normalised using Knights and Ruiz matrix balancing, considering only the genomic region chr13: 54,000,001–57,300,000 (length 3,300,000 bp), and exported at 5 kb resolution[46–48]. All maps were produced on the wildtype reference genome in order to use the same genomic coordinates across all samples Subtraction maps of the KR normalised maps were scaled together across their sub-diagonals to normalise for distance-dependant signal decay. Here, each subdiagonal vector in a matrix is divided by its sum and multiplied by the average of the sums in the two matrices. All maps were visualised as heatmaps where values above the 99th percentile were truncated for visualisation purposes. For further details can be found in ref. 4. The codes are available on https://github.com/bompadreolimpia/Bompadre_etal_2024.

**Single Cell RNA-Seq**. Sequenced reads were mapped to the custom genome mm39_dCas9P300 and corresponding GTF file using the 10X Genomics Cell Ranger 6.1.2 software. Data filtering (nFeature_RNA > 200 & nFeature_RNA < 5000 & percent.mt <5 & nCount_RNA > 1000 & nCount_RNA < 26000), quality control, normalization, scaling, dimensional reduction, and doublet identification were performed using Seurat 4.3.0 and DoubletFinder 2.0.3. Cells were further filtered to exclude blood cells present in our dataset (percent.mt > 1 & percent.mt <5).

*Merging and Normalization*. Following individual dataset filtering and normalization, the two wildtype forelimb and hindlimb replicates and the two *Pitx1^{Inv1/+}* forelimb replicates were merged as single Seurat objects. To account for potential variance due to cell-cycle variations, cell cycle regression was implemented using the CellCycleScoring method with a predetermined list of marker genes (Tirosh et al., 2016). The dataset underwent additional normalization through SCTransform with standard parameters, incorporating the scored cell-cycle and the dCas9P300 feature as regressed variables (Hafemeister et al., 2019).

*Clustering of Whole Limbs and Mesenchyme*. The cells were clustered after cell cycle and dCas9P300 regression using the SCTransform Seurat package. For clustering, PCA (50 npcs) and UMAP (50 dims) were utilized, and the closest neighbors of each cell were calculated. The Seurat FindClusters function was employed with a resolution of 0.1, defining 9 clusters. Cluster identification was performed with the FindMarkers function, enabling the selection of differently expressed gene markers among clusters (ident.1, only.pos=TRUE). Three mesenchymal cell clusters and two epithelial cell clusters were merged, with 6 final clusters remaining, where the FindMarkers function was re-run (Supplementary Data 4).

Given the exclusive expression of *Pitx1* and *Shox2* in the mesenchymal cells of the limb, downstream analysis focused on these populations. The 3 mesenchymal cell populations were merged and reclustered. PCA of 20 npcs and UMAP of 20 dims were applied, and closest neighbours were calculated for each cell. Using Seurat FindClusters, 10 clusters were defined with a resolution of 0.3. FindMarkers was then run for each cluster, selecting gene markers (ident.1, only.pos=TRUE). Two Irregular Connective Tissue clusters were merged and FindMarkers was re-run on the final 9 clusters (Supplementary Data 4). Analysis of *Pitx1* and dCas9P300 expression in wildtype and *Shox2^{dCas9P300/+};ColA1^{TSSsgR/+}* forelimbs was restricted to these limbs only which were subset after cluster analysis. UMAP density plots were obtained using the R package *Nebulosa* v1.8.0 and *scTransform*

v0.4.1. The codes are available on https://github.com/bompadreolimpia/Bompadre_etal_2024.

*Expression correlation*. To calculate the correlation of expression of two genes in a sample from single-cell-RNAseq data we employed *baredSC* v2.0.0 (Lopez-Delisle, et al., 2022). Here, the confidence interval of correlation is given as a percentage and the *p*-value, where *p* is the probability for the correlation coefficient to be negative, is the mean probability with the estimated standard deviation of this mean probability. The code is available on https://github.com/bompadreolimpia/Bompadre_etal_2024.

### Reporting summary
Further information on research design is available in the Nature Portfolio Reporting Summary linked to this article.

## Data availability
RNA-seq, C-HiC, ChIP-seq sequencing data are available in the GEO repository under the accession number GSE259212: https://www.ncbi.nlm.nih.gov/geo/query/acc.cgi?acc=GSE259212. Source data are provided with this paper Source data are provided with this paper.

## Code availability
RNA-seq, scRNA-seq, ChIP-seq and Capture-HiC processing codes are available here: https://github.com/bompadreolimpia/Bompadre_etal_2024 and citable via https://doi.org/10.5281/zenodo.15269742. GRCm39/mm39_eGFP-SV40pA sequence and the filtered GTF file are available on Zenodo: https://doi.org/10.5281/zenodo.7837435. GRCm39/mm39_dCas9-P300 sequence and the filtered GTF file are available on Zenodo: https://zenodo.org/records/11122106.

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

## Acknowledgements

We thank Mylène Docquier, Brice Petit, Didier Chollet and Christelle Barraclough from the iGE3 sequencing facility. We thank Grégory Schneiter, Lan Tran and Cécile Gameiro from the Flow Cytometry facility. We thank Olivier Fazio, Angélique Vincent and Fabrizio Thorel from the Transgenic facility. We thank Lucille Delisle for bioinformatic support. The computations were performed at University of Geneva using Baobab HPC service. This study was supported by grants from the Swiss National Science Foundation PP00P3_176802, PP00P3_210996, 320030-231203 to G.A., from the Novartis and Boninchi Foundations to G.A. M.K. lab is supported by the Canadian Institutes of Health Research grant CIHR 174989.

## Author contributions

G.A. conceived the project. O.B. and R.R.G. performed scRNA-seq preparations and analysis. O.B. and F.D. targeted and characterised the dCas9-P300 activator mESC clones and embryos. O.B. and A.R. performed mESC targetings, prepared the cells for tetraploid aggregation and performed WISH and skeletal preparations. O.B. performed embryo imaging, ChIP-seq, C-HiC and RNA-seq and analyses. M.K., F.G.-M. and C.G. provided the Eed knock out and control tissues for Capture-HiC. G.A. and O.B. wrote the manuscript with input from the remaining authors.

## Competing interests

The authors declare no competing interests.
