## [Transparent Peer Review file · Nature Communications]

Liebenberg syndrome severity arises from variations in Pitx1 locus topology and proportion of ectopically transcribing cells

Corresponding Author: Professor Guillaume Andrey

Version 0:

Reviewer comments:

Reviewer #1

(Remarks to the Author)

This new version of the manuscript addresses most of my concerns and I believe it is now ready for publication.

Reviewer #2

(Remarks to the Author)

First of all, I would like to thank the authors for their efforts to address my previous concerns. As I already mentioned in my previous revision, the presented work is extensive and of high technical quality and the results are in general clear and robust. My main concern relates to the conceptual novelty of the work regarding how changes in the linear distances between enhancers and genes can lead to changes in gene expression levels and bursting frequency without necessarily having a major impact in enhancer-gene contacts. Nevertheless, the work is still highly relevant as it showcases how changes in linear distances between genes and cognate (rather than ectopic) enhancers can lead to medically relevant phenotypic defects.

I only have the following suggestion: The manuscript would benefit from including a Figure in which at least some of the SVs identified in human patients are actually depicted and compared with the re-arrangements generated in mice.

Reviewer #3

(Remarks to the Author)

The editor has just provided the authors' response to the reviewers, which I previously did not have access to. Their replies are clear and reasonable, including the points regarding HoxC. They have addressed all of my comments, and I support the publication of the manuscript.

Reviewer #1

This new version of the manuscript addresses most of my concerns and I believe it is now ready for publication.

We thank this reviewer for his/her helpful comments and suggestions to our manuscript.

Reviewer #2

First of all, I would like to thank the authors for their efforts to address my previous concerns. As I already mentioned in my previous revision, the presented work is extensive and of high technical quality and the results are in general clear and robust. My main concern relates to the conceptual novelty of the work regarding how changes in the linear distances between enhancers and genes can lead to changes in gene expression levels and bursting frequency without necessarily having a major impact in enhancer-gene contacts. Nevertheless, the work is still highly relevant as it showcases how changes in linear distances between genes and cognate (rather than ectopic) enhancers can lead to medically relevant phenotypic defects.

We appreciate this reviewer's insightful comments and suggestions on our manuscript.

I only have the following suggestion: The manuscript would benefit from including a Figure in which at least some of the SVs identified in human patients are actually depicted and compared with the re-arrangements generated in mice.

We have produced a new Figure S9 (see below) and referred to in the discussion line 371

Reviewer #3

The editor has just provided the authors' response to the reviewers, which I previously did not have access to. Their replies are clear and reasonable, including the points regarding HoxC. They have addressed all of my comments, and I support the publication of the manuscript.

We thank this reviewer for his/her helpful comments to our manuscript.